# Coseismic displacements of the 14 November 2016 Mw7.8 Kaikoura, New Zealand, earthquake using the Planet optical cubesat constellation

Andreas Kääb[1], Bas Altena[1], Joseph Mascaro[2]

[1]Department of Geosciences, University of Oslo, Oslo, 0316, Norway
[2]Planet, San Francisco, postal code 94103, USA

*Correspondence to*: Andreas Kääb (kaeaeb@geo.uio.no)

Received:  – Published in Nat. Hazards Earth Syst. Sci. Discuss.:
Accepted:  – Published:

**Abstract.** Satellite measurements of coseismic displacements are typically based on Synthetic Aperture Radar (SAR) interferometry or amplitude tracking, or based on optical data such as from Landsat, Sentinel-2, SPOT, ASTER, very-high resolution satellites, or airphotos. Here, we evaluate a new class of optical satellite images for this purpose – data from cubesats. More specific, we investigate the PlanetScope cubesat constellation for horizontal surface displacements by the 14 November 2016 Mw7.8 Kaikoura, New Zealand, earthquake. Single PlanetScope scenes are 2-4 m resolution visible and near-infrared frame images of approximately 20-30 km × 9-15 km in size, acquired in continuous sequence along an orbit of approximately 375-475 km height. From single scenes or mosaics from before and after the earthquake we observe surface displacements of up to almost 10 m and estimate matching accuracies from PlanetScope data between ±0.25and ±0.7 pixels (~ ±0.75 to ±2.0 m), depending on time interval and image product type. Thereby, the most optimistic accuracy estimate of ±0.25 pixels might actually be typical for the final, sun-synchronous and near-polar-orbit PlanetScope constellation when unrectified data are used for matching. This accuracy, the daily revisit anticipated for the PlanetScope constellation for the entire land surface of Earth, and a number of other features, together offer new possibilities for investigating coseismic and other Earth surface displacements and managing related hazards and disasters, and complement existing SAR and optical methods. For comparison and for a better regional overview we also match the coseismic displacements by the 2016 Kaikoura earthquake using Landsat8 and Sentinel-2 data.

## 1 Introduction

Coseismic displacements are typically measured from satellite synthetic aperture radar (SAR) data using radar interferometry or radar tracking techniques (Massonnet and Feigl, 1998; Michel et al., 1999; Avouac et al., 2015; Kargel et al., 2016, and many others). These data and methods have the advantage to cover large areas at once (for instance, Sentinel-1 swath width is ~250 km for interferometric wide swath mode), independent of cloud cover and solar illumination, and enable

displacement accuracies in the range of centimetres if interferometric phase coherence is preserved. The interferometric measurements reveal the displacement component in line-of-sight from the radar satellites. Radar tracking methods measure the azimuth (flight direction of satellite) and range (line-of-sight) components of the displacements with, roughly, metre-accuracy for entire image areas, and potentially better for selected strong artificial or natural radar reflectors (Michel and Rignot, 1999; Singleton et al., 2014; Wang and Jonsson, 2015). Both methods can be combined (e.g., Fialko et al., 2001). Typical revisit times for current radar satellites are on the order of a few days to weeks (e.g., 6 days for the Sentinel-1 constellation of two satellites; 14 days for ALOS-2 PALSAR; 11 days for TerraSAR-X; 24 days for Radarsat-2).

Repeat optical satellite data are significantly less used for matching coseismic displacements, among others due to their sensitivity to cloud cover and their reduced accuracy compared to radar interferometry. If suitable data are available, though, optical images can typically be matched with higher accuracy than radar data of similar spatial resolution because SAR data are affected by speckle noise, which is more sensitive to ground changes than repeat optical data. Furthermore, optical data can be a more independent displacement measurement, as radar interferometry involves phase ambiguity that can be difficult to solve when displacement gradients are large or complex.

Coseismic displacements have, for instance, been measured on repeat data from Landsat (Liu et al., 2006; Avouac et al., 2014; Barnhart et al., 2014), ASTER (Avouac et al., 2006), SPOT (Dominguez et al., 2003; Leprince et al., 2007; Konca et al., 2010), very high resolution optical satellites (Barnhart et al., 2015; Zhou et al., 2015), or air photos (Michel and Avouac, 2006; Ayoub et al., 2009). Coseismic displacements from Sentinel-2 data have to our best knowledge not yet been published in peer-reviewed journal publications, but are used by operational services (COMET, 2016). Landsat (16 day repeat orbit, 15-30 m resolution), ASTER (16 day repeat orbit, 15 m Visible and Near Infrared, VNIR, resolution) and Sentinel-2 (10 day repeat orbit, 5 day repeat orbit once the Sentinel-2A and 2B constellation is complete, 10-20 m resolution depending on band) are useful for regional displacement-fields and provide approximately the horizontal motion components due to their nadir-looking geometry (only ASTER is occasionally pointed in cross-track direction). Landsat and Sentinel-2 data are provided only as orthorectified version (ASTER optionally) so that positions in these orthoimages are potentially contaminated by cross-track distortions that propagated from errors in the DEM used for orthorectification (Kääb et al., 2016; Altena and Kääb, 2017). Avouac et al. (2006) and Girod et al. (2015) demonstrated refined sensor models for ASTER that reduce georeference noise significantly, and Avouac et al. (2006) developed this approach further to enable measurement of coseismic displacements from ASTER data with an accuracy of few metres.

Due to their high spatial resolution of up to 30 cm, repeat data from very high resolution optical satellites such as the WorldView series or Pleiades can be used to measure coseismic displacements with centimetre to decimetre accuracy (Barnhart et al., 2015; Zhou et al., 2015). Typically, however, these satellites provide no regular acquisitions, and tasked acquisitions can be quite oblique.

Global Navigation Satellite System (GNSS) measurements, such as from GPS, of coseismic and other tectonic movements provide mm to cm precise 3-dimensional displacements in a global reference system on selected points were permanent stations are running. Such high-precision point measurements can thus be highly synergistic to less precise, but area-wide

satellite displacement measurements, for instance by providing absolute georeference to else only relative satellite measurements.

In this contribution we evaluate a new class of optical satellite data to estimate coseismic displacements – optical cubesats. As a test case we investigate lateral ground displacements associated with the 14 November 2016 New Zealand earthquake. This magnitude 7.8 Mw earthquake occurred in the first few minutes of 14 November 2016 at a depth of approximately 15 km in the northeast of the South Island of New Zealand, near the town of Kaikoura, and was in terms of magnitude the second strongest earthquake in New Zealand since European settlement (GeoNET, 2016; USGS, 2016). Surface motion happened mainly at the Kekerengu Fault, Papatea/Waipapa Bay Fault, Hundalee Fault and Hope Fault, which all are part of a fault system between the Australian and Pacific Plates (GeoNET, 2016) (Fig. 1). Media images from after the earthquake show significant surface ruptures at the above faults with vertical and horizontal motion clearly visible (GeoNET, 2016). A number of landslides were obviously triggered by the earthquake, and in some areas the sea bed was lifted several meters (GeoNET, 2016; Sciencealert, 2016).

In this paper we assess the potential and limitations of optical cubesats, and investigate to which extent they can complement the above-mentioned established radar and optical data and methods. For this purpose we focus in particular on the cubesat constellation by the company Planet. First, we describe the Planet cubesat constellation and details of the image matching methods used in this study. Next, we present the results and discuss their performance and characteristics in order to evaluate the usefulness for coseismic displacements. In the final conclusions we try to answer the research questions raised at the start of this paragraph.

## 2 The Planet cubesat constellation

The Planet cubesats (cubesats are sometimes also referred to as nanosatellites), called PlanetScope or more popular 'doves' and with single cubesat series called 'flocks', have a size of about 10 cm x 10 cm x 30 cm, i.e. are 3-unit cubesats (one cubesat unit is 10 cm x 10 cm x 10 cm). Their main component is a telescope and CCD area array sensor, complemented by solar panels for power generation, a GNSS receiver for satellite position, a startracker for satellite orientation, reaction wheels for attitude control and stabilisation, an antenna for down- and uplink, batteries and on-board storage. One half of the $6600 \times 4400$ pixel CCD array acquires red-green-blue data and the other half NIR, both in 12 bit radiometric resolution. The PlanetScope satellites provide images of about 2-4 m spatial resolution, and a size of individual scenes of roughly 20-30 km $\times$ 9-15 km (Planet Team, 2016) (Marshall and Boshuizen, 2013; Boshuizen et al., 2014; Foster et al., 2015). Ground resolution and scene size vary with flying height and satellite version. While most other optical Earth observation instruments in space deliver images in pushbroom geometry (i.e. linear sensor arrays scanning the swath width in orbit direction), the data from the Planet satellites are frame images – an important detail with respect to systematic distortions within the image product. That is, each complete scene is taken at one single point in time, has one single acquisition position and one single bundle of projection rays. For comparison, pushbroom sensors integrate an image over a certain time

interval so that acquisition time, position and attitude angles vary throughout an image, which may lead to higher-order image distortions (Nuth and Kääb, 2011; Kääb et al., 2013; Girod et al., 2015) .

In its final stage, the Planet cubesat constellation will consist of around 120 cubesats following each other in one near-polar orbit of 96 degree inclination and an altitude of about 475 km (Fig. 2). The distance of the cubesats to each other in this orbit

is designed in a way so that the longitudinal progression between them over the rotating Earth leads to a void-less scan of the surface (except the polar hole) and the full constellation provides sun-synchronous coverage of the entire Earth with daily resolution (Fig. 1). At the time where the analyses were done for the present study about 60 Planet cubesats were in space, with the majority of them not yet in a final near-polar orbit but in an International Space Station (ISS) orbit of 52 degrees inclination and ~375 km height. As one consequence for this study, the constellation did not yet provide daily global

coverage and the images are taken at varying times of the day and with varying azimuths. However, 88 more PlanetScope cubesats were successfully launched on 14 February 2017 into the final sun-synchronous near-polar orbit. These cubesats should be operational within a few weeks to months after the time of this writing and thus the PlanetScope constellation be complete. We anticipate that the doves in sun-synchronous orbit will function for 3-5 years. Planet plans to keep the constellation complete by continuously supplying new satellites.

For image matching purposes the geometric characteristics of repeat imagery is of particular interest and will thus be discussed in more detail in the following. PlanetScope images are available in different processing versions, and here we use 'unrectified' and 'analytic' data, both accessible from Planet. 'Unrectified' data come with minimal radiometric processing and are in the original frame geometry, i.e. central projection. 'Analytic' data are radiometrically processed and orthorectified. Radiometric calibration is done through a mixture of pre-launch calibration, calibration sites, and calibration

during an image co-registration process to other satellite images (the latter described below). The current lens model used during georectification was estimated once for all telescopes of the current building series and is accurate within a fraction of a pixel, better than 0.1 pixels. The image orientation parameters from on-board measurements are refined by matching the scenes onto other orthorectified images and the PlanetScope scenes are then orthoprojected using a DEM. For the first step, coregistration, Planet uses the "best available" reference images for a ground reference raster. For example, national airphoto

mosaics, ALOS PRISM, RapidEye, and then Landsat 8 data are preferentially used, respectively, depending on which data are available and give sufficient matches. The orthorectification uses a "best available" DEM depending on location. All these processing steps and data are constantly assessed and updated, and if appropriate the archive reprocessed.

The image orientation parameters from on-board measurements are refined by matching the scenes onto a Landsat mosaic and the images are orthoprojected using a DEM. As for all orthoprojected satellite data, vertical errors in the

orthorectification DEM lead to lateral distortions in the resulting PlanetScope orthoimages, the size of which is proportional to the DEM error and the off-nadir viewing angle. For instance, for an orbit height of 400 km and a perfect nadir image of 20 km swath width (typical parameters for PlanetScope images), i.e. a maximum off-nadir distance of 10 km, a DEM error of 15 m (a typical accuracy for the SRTM DEM) (Nuth and Kääb, 2011) will translate to a maximum orthorectification distortion at the image margins of 38 cm. The Planet cubesats are controlled to acquire data within an off-nadir angle of ±2º,

which translates for an orbit height of 400 km to a maximum off-nadir offset on the ground of 14 km in image centre and 24 km at its margin. For this maximum off-nadir viewing the orthorectification offsets resulting from a vertical DEM error of 15 m are 52 cm in the image centre and 90 cm at the image margin. For an orbit height of 475 km and a scene width of 30 km the latter offset numbers get 44 cm in the scene centre and 94 cm. Both scenarios represent the worst case for the propagation of orthrectification DEM errors into lateral distortions in PlanetScope images.

These expected orthorectification distortions are likely smaller than potential georeferencing errors from imperfect satellite positions and attitude angles, and their refinement from registering the images to reference images. Current pointing error for the satellites is on the order of 5 km prior to georectification. After georectification the georeference accuracy is 10 m RMSE according to specifications, and 6.5 m, i.e. better than the specifications, according to validation measurements by Planet. At the time of writing the image referencing procedure is being upgraded, though.

However, distortions between unrectified frame images due to errors in image orientations are a standard problem in stereo-photogrammetry, called relative orientation. Such distortions are of analytical nature and can thus in principle be modelled and removed — in contrast to distortions from orthorectification DEM errors that are largely of unpredictable nature, depending on DEM errors. The fact that Planet images are frame images and are also available in unrectified form opens therefore in theory possibilities for own orthorectification or modelling of georeferencing errors to increase the accuracy of displacements matched from repeat images.

It should also be noted that orthorectification DEMs (or DEMs for topographic phase removal within SAR interferometry) are by necessity outdated unless acquired simultaneously with image acquisition (Stumpf et al., 2014). Any orthorectification, no matter how accurate in space, is therefore temporally corrupted by the fact that the ground is a moving target, always changing in time. Typically, ground changes will be small enough to not have significant effect on orthorectification, but for instance for landslides, major earthquakes, or glaciers the resulting offsets are an inherent problem of orthorectification of monoscopic data (Kääb et al., 2016; Altena and Kääb, 2017). The small field of view of PlanetScope cubesats and the resulting small sensitivity to topographic distortions, the frame geometry of the PlanetScope cameras, and the accessibility of unrectified images all contribute to minimize and potentially remove topographic distortions.

## 3 Data and methods

To investigate coseismic displacements from repeat optical data we match images from before and after the 14 November 2016 earthquake over the north-eastern section of the southern island of New Zealand. In order to get a regional overview of displacements we first match Sentinel-2 data of 3 October and 5 December 2016 (NIR band 8, 10 m resolution; 63 days), and the closest suitable Landsat 8 data around the earthquake date from 12 October and 15 December 2016 (pan band 8, 15 m resolution; 64 days; Fig. 1). For detailed displacements over main ruptures we select PlanetScope images of 27 October, 21 November and 28 November 2016 (i.e. 25 and 32-day pairs; Fig. 1). A number of other suitable Sentinel-2 and

PlanetScope images are available, too, but the selected ones seemed best to us in terms of illumination, cloud cover and proximity to the earthquake date.

In order to cross-check the potential displacement accuracy from PlanetScope data, we also measured displacements from two PlanetScope scenes of 20 and 25 November 2016 just to the southwest outside of the section shown in Fig. 1. These images stem from a sun-synchronous near-polar repeat orbit such as to be expected as standard from the final Planet constellation — and occasionally already provided at the time of writing from the preparatory constellation. No such type of scenes from sun-synchronous near-polar orbits was available directly over the section of Fig. 1 around the earthquake date so that we use Planet scenes acquired from preliminary ISS-type orbits over the region of Fig 1. Daily MODIS data around the earthquake date show suitable imaging conditions on 1, 3, 8 and then again on 15, 18, 19 and 21 November etc. where the final sun-synchronous daily Planet imaging constellation would thus have had acquired data. The above test with data from sun-synchronous near-polar obits and with 5 days interval between scenes (20 and 25 November) seems thus representative and realistic.

For matching the repeat Sentinel-2, Landsat 8 and PlanetScope data we use standard normalized cross-correlation (NCC), solving the cross-correlation in the spatial domain and reaching sub-pixel accuracy by interpolation of the image (Kääb and Vollmer, 2000; Debella-Gilo and Kääb, 2011a; Kääb, 2014). The matching window sizes used for the Sentinel-2 data were 20×20 pixels (200 m), for Landsat 8 15×15 pixels (225 m), and for PlanetScope 20×20 pixels (60 m). Tests with different window sizes are not the focus of this study (Debella-Gilo and Kääb, 2011b). Measurements with a correlation coefficient smaller 0.7 are removed and no other post-processing is applied. Offset patterns such as global offsets, jitter or stripes, which might have a magnitude of several metres for Landsat 8 and Sentinel-2 (Kääb et al., 2016), have not been investigated and corrected. The offsets presented here are thus relative between the matched scenes and not necessary absolute offsets in some global reference system.

Preservation of absolute georeference over the earthquake is tricky as we cannot be sure about a change in position of the plates involved from our satellite data alone. The pointing accuracy of the satellites used is not accurate enough for that purpose and co-registration steps are involved in the processing of the Landsat and PlanetScope data anyway (and in the near future also for Sentinel-2). The focus of our evaluation lies therefore on relative displacements between scene zones. Such strain maps are also produced when (In)SAR techniques are used. Absolute georeference problems could be reduced by co-registering PlanetScope data to selected images and image sections of, for instance, Landsat 8 or Sentinel-2 data, or airphoto orthoimage mosaics. Also GNSS measurements of coseismic displacements could be used fit PlanetScope-derived displacements onto.

**4 Results**

**4.1 Planet, Sentinel 2 and Landsat 8 coseismic displacements**

Figure 1 shows the horizontal coseismic displacements from the Sentinel-2 data of 3 October and 5 December (Fig. 1, upper row), and from the Landsat 8 data of 12 October and 15 December 2016 (middle row). The main rupture by the earthquake
along the Kekerengu fault has an azimuth of very roughly 45º and we thus transform the measured displacements to a Cartesian coordinate system rotated by 45º, i.e. we show the SW-NE (Fig. 1, left column) and NW-SE (right column) displacement components instead of W-E and S-N. From the repeat Sentinel-2 and Landsat 8 data the main rupture is along a sharp line over the Kekerengu fault. There, we find relative displacements of around 9 m with an azimuth of roughly 65°. At the Papatea fault we obtain relative displacements of around 6.5 m with an azimuth of roughly 130°.

To evaluate PlanetScope data we match a two-scene mosaic of 28 November 2016 with a mosaic of four scenes of 27 October over parts of the Kekerengu and Papatea fault ruptures (Fig. 1, rectangle A) and show the W-E and S-N components of the displacements obtained (Fig. 3, middle row). Both mosaics have been compiled from standard orthorectified PlanetScope products as is, without any additional own corrections or adjustments. All images used for the mosaics were available with the same ground resolution so that no resampling was necessary beforehand matching them. The measured
displacements show a sharp rupture over the Kekerengu fault of around 6 m with a rupture azimuth aligning well with the azimuth of the displacement. Over the Papatea fault the rupture is less straight and rather oblique to the horizontal displacement of about 5.5 m. The latter displacement agrees well within the errors bounds with the Sentinel-2 results. The displacement field derived from the PlanetScope data is very dense and shows details that become not obvious from Sentinel-2 and Landsat 8, for instance the higher W-E displacements in the southernmost zone of the section in Fig. 3.
Between the Kekerengu and Papatea fault ruptures in Fig. 3 we observe gradients in both the W-E and S-N displacement components resulting from an increase of displacement magnitude towards the Papatea fault rupture accompanied by a rotation of the displacement field towards east closer to the rupture (Fig. 3, top row).

The lower panel in Fig. 3 shows Sentinel-2 derived displacements for comparison, i.e. details of Fig. 1 (upper row) with, however, N-S and E-W displacement components.

Over another section at the Kerengu fault rupture (Fig. 1, rectangle B) we match a PlanetScope scene of 21 November with the 27 October mosaic (Fig. 4). Only the W-E components of displacements are shown as the S-N ones look very similar. The measurements show a sharp displacement over the rupture of around 8.5 m with an azimuth slightly oblique to the rupture. Again, the displacement from PlanetScope data agrees well within error bounds with the Sentinel-2 results of 9 m. Over the Clarence River flood plain no measurements are possible. The lower panel in Fig. 4 shows Sentinel-2 derived
displacements over the same section for comparison.

Figure 5 shows a small detail of Fig. 3 (C in Fig. 3) with the 27 October – 28 November 2016 PlanetScope-derived displacements, once with the 27 October, once with the 28 November image in the background. At this location, the seabed was lifted up by roughly two meters east of the rupture that is also well visible in the 28 November image (GeoNET, 2016;

Sciencealert, 2016). The main rupture obtained from the displacements is offset from the seabed rupture visible in the images.

Figure 6 shows a small detail of Fig. 4 with the PlanetScope image of 21 November in the background (location D in Fig. 4). Matches did not achieve correlation coefficients larger than 0.7 over the rupture itself due to high deformations and surface

destruction, and are thus removed. At these places the rupture is visible in the underlying Planet image, confirming the accurate delineation of the rupture by the derived displacements.

Figure 7 (location E in Fig. 4) shows a detail of Fig. 4 with the PlanetScope images from 27 October and 21 November behind the displacements. Here, presumable vertical uplift of the terrain to the southeast, accompanying the horizontal displacements by the rupture, have dammed up Clarence River and changed its course as visible in the PlanetScope images.

Figure 8 (location F in Fig. 4) illustrates landslides due to the 14 November earthquake close to the Kekerengu fault rupture in order to give an impression of the visual characteristics of the PlanetScope data and other uses of the data related to earthquake disaster management. To the southeast in the figure, the rupture is well visible as a bright line.

## 4.2 Planet data stable ground test

As the Planet constellation was not yet final at the time of the 2016 New Zealand earthquake no images were available from

the sun-synchronous near-polar orbit close to the earthquake date. To simulate displacement measurements based on PlanetScope data of this final constellation we match the overlap of PlanetScope images near our study site from 20 and 25 November 2016. Both scenes come from the sun-synchronous near-polar orbit (Fig. 9). The type of terrain and land cover over these scenes is very similar to the ones applied above over the ruptures. We perform three assessments:

Figure 9b: Matching the orthorectified versions of both images shows a mean offset of only 0.25 m, i.e. less than 0.1 pixels.

The standard deviation of this offset, that is the variability of the individual displacements, is around 1.9 m, and the mean magnitude of the individual displacements is 1.6 m. This indicates an accuracy of individual displacements of about ±0.6 pixels.

Figure 9c: We use the unrectified versions of the two scenes, co-register them using a $1^{st}$-order polynomial (i.e. removing a global shift and approximately a rotation), and match them. Over most of the overlap we obtain a standard deviation of

displacements of around 0.2-0.3 pixels (~0.75 m). Towards the left and right margins we see distortions between the scenes of up to 5-6 pixels. These are due to the superposition of the lens and image distortions of both images, distortions that are not corrected for in the unrectified data version and not sufficiently reduced by our simple $1^{st}$-order polynomial co-registration. Comparison with the matching based on the orthorectified images versions (Fig. 9b) shows that these effects are efficiently removed during the processing steps by Planet towards orthorectified data.

Figure 9d: We use the same procedure as for the results Fig. 9c, but use a $2^{nd}$-order polynomial instead, i.e. including quadratic terms in the co-registration. Now, the distortions to the right and left margins are mostly removed and a pattern of undulations of ±0.1 pixel in amplitude becomes visible. This pattern is also present in the test of Fig. 9c but difficult to visualize there due to the overlying and much larger global scene distortion. We assume this undulating pattern is a

superposition of higher-order distortions in the individual images. Again, we cannot find such pattern anymore between the orthorectified scenes. Like in the test Fig. 9c, also in Fig. 9d the standard deviation of individual displacements is on the order of 0.2-0.3 pixels.

## 5 Discussion

In general, the Landsat 8 and Sentinel-2 results are similar. Also for a number of details in the displacement field both agree, but there are also some minor differences. The latter could easily be due to imperfect co-registration within the matching pairs, or deviations/distortions of absolute georeference between the matching pairs (see end of above Sections 2 and 3). Overall, the displacement field from the Sentinel-2 data seems slightly sharper and with less outliers compared to Landsat 8, as expected for the higher image resolution of Sentinel-2. For optimal ground conditions (e.g. flat desert) repeat Sentinel-2 data can be matched with an accuracy of up to 0.1-0.2 pixels (1-2 m) for single displacements (Kääb et al., 2016). From the standard deviation of displacements over homogenously displacing image sections we estimate in this study a relative accuracy for individual displacements of about ± 0.4 pixels (4 m) for Sentinel-2 and about ± 0.25 pixels (3.8 m) for Landsat 8, for the matching window sizes, ground conditions and time interval specific to our study. The differences between the Sentinel-2 and Landsat 8 derived displacements are on average -0.8 ± 9.2 m in SW-NE and -1.5 ±4.2 m in NW-SE (Sentinel 2 minus Landsat 8). The maps of differences (not shown) display a smoothly undulating pattern that could roughly be connected to topography, pointing to terrain-correction differences as a possible source of the differences between Sentinel-2 and Landsat 8 displacements (Kääb et al., 2016). Further in-depth investigations of the Sentinel-2 versus Landsat 8 differences are outside the scope of this paper.

Whereas the overall displacement pattern between Sentinel-2 and PlanetScope agrees, the Sentinel-2 displacements show more noise and outliers (Figs. 3 and 4, lower rows), which gives in parts the impression of larger displacement magnitudes. The average difference between both displacement fields is 5.3 ± 5.2 m (vector magnitude) for region A, and 4.2 ± 5.2 m for region B, whereby the 5.3 m or 4.2 m offset, respectively, reflect the lacking and thus imperfect co-registration of both data sets. The ±5.2 m relative uncertainty (1 σ) of displacements should mainly stem from the Sentinel-2 derived ones, as Figs. 3 and 4 suggest. The difference maps between the PlanetScope and Sentinel-2 displacements for the sections of both Figs. 3 and 4 (not shown) display mostly noise, but also some patterns potentially related to topography and thus orthorectification. There seems to be also some difference on the order of 2 m in overall displacement on either side of, for instance, the Papatea fault (cf. Fig. 3), the reason of which is open to us (influence of shadow changes, orthorectification artifacts, ?).

From the standard deviation of displacements over homogenously moving sections of the scenes used here (Figs 3 and 4), we estimate a relative accuracy of individual displacements of about ± 0.67 pixels (2 m) for the PlanetScope data and the matching window sizes, ground conditions and time intervals specific to our study.

Remarkably, neither in the matching between the orthorectified scenes (Figs. 3, 4 and 9b) nor between the unrectified scenes (Fig. 9c and d) notable topographic effects become visible. This confirms that, due to the small field of view of the Planet

satellites and their nadir looking, the image geometry is quite insensitive to orthorectification DEM errors or topographic distortions, respectively. In consequence, unrectified PlanetScope scenes could in a number of applications be used directly for displacement measurement without applying any topographic correction or DEM-based orthorectification, respectively.

Overall, our measurement of coseismic displacements using PlanetScope data and the test over stable ground suggest a
relative accuracy of around ± 0.6 - 2.0 m (0.2 - 0.7 pixels; 1 standard deviation) for individual displacements. Important, when averaging such displacements over defined zones, as one would typically do for investigating coseismic displacements, the accuracy (standard error) of a resulting mean zonal displacement will be significantly better, depending on the number of displacements averaged and their dependency to each other.

## 6 Conclusions

We demonstrate horizontal coseismic displacements of the 14 November 2016 Kaikoura, New Zealand, earthquake from repeat Sentinel-2, Landsat 8 and PlanetScope data. Over the two faults investigated we find horizontal surface slip of around 6-9 m. The main goal of this study was to assess the potential of PlanetScope data for this purpose.

The main limitation of optical data in general is their dependency on cloud-free conditions and solar illumination, in contrast to SAR acquisitions. Also, due to their nadir-viewing geometry most optical data give no access to the vertical component of
coseismic (and other) terrain displacements. Where phase coherence is preserved within SAR radar data, displacement can through interferometry be measured by a precision that can seldom be matched by optical remote sensing data. However, where this phase coherence is not given, optical data can be a valuable alternative to radar data for coseismic (and other Earth surface) displacement measurements. The estimated displacements can also be of help to better unwrap SAR interferometry data. When the gradient of the strain increases too much, the interferometric phase fringes are difficult to
follow (unwrap). However, displacements from image matching are not ambiguous so that two-dimensional integration is not needed.

One of the main advantages of PlanetScope data for coseismic displacements is their anticipated daily repeat. This maximizes the chances to receive cloud-free images and to cover unexpected events such as earthquakes. The according small time periods of a few days that form the image matching pairs and the related small changes in ground and
illumination conditions, together with the frame geometry of the PlanetScope images, enables relative measurement accuracies of as low as ±0.2-0.3 pixels (~ ±0.75 m) for individual displacements, and potentially much better for zonal averages. In detail, we performed three tests about this potential displacement accuracy: variance of displacements over homogenously displacing areas using orthorectified PlanetScope images from preliminary orbits (not sun-synchronous, not near polar) over actual ruptures (obtaining ±0.67 pixels); variance of displacements over stable terrain using orthorectified
PlanetScope images from final orbits (sun-synchronous, near polar) with 5-day repeat (±0.63 pixels); variance of displacements over stable terrain using unrectified PlanetScope images from final orbits (sun-synchronous, near polar) with 5-day repeat (±0.2-0.3 pixels). In combination with the high spatial image resolution of around 3 m, details in the

displacement field can thus become apparent that are not detected in Sentinel-2 or Landsat 8 data. The envisaged daily repeat by PlanetScope data will further improve the above displacement accuracy by enabling to measure displacements in several image pair combinations and thus exploiting a temporal stack of images and displacements (Dehecq et al., 2015; Kääb et al., 2016; Altena and Kääb, 2017; Stumpf et al., 2017).

In comparison to Sentinel-2 and Landsat 8, a main limitation of PlanetScope scenes can be their small extent of only a few 100 $km^2$. Precise georeferencing between images before and after large-scale coseismic displacements can thus be difficult as all terrain covered by a scene might have been displaced or deformed in some way. In such cases, the data provide relative displacements over smaller areas or well-defined ruptures, i.e. strong gradients in a displacement field. Long-wavelength variations or low gradients in a displacement field will be more complicated to measure as these cannot easily be

discriminated from distortions in the repeat images or their co-registration. The above problems can be in parts reduced by mosaicking longer stripes of scenes instead of using single scenes, as demonstrated in our study for the 27 October and 28 November data. Finally, the above matching accuracy of on the order of ±1 m will prevent detecting small (coseismic) displacements.

Though listed above as disadvantage, the small PlanetScope scene size and the connected small field-of-view, together with

their nadir acquisition, have on the other hand the advantage that topographic distortions in PlanetScope data are small and the resulting orthoimages quite robust against vertical errors in the DEM used for orthorectification. This effect contributes also to the good matching results above.

Finally, even if not the main focus of this study on coseismic displacements, the visual information provided by the high resolution, daily repeat PlanetScope data can be very valuable for mapping and managing the impacts of earthquakes, such

as ruptures, landslides, damming of rivers, damaged infrastructure, etc. Because the downlink network of the Planet constellation has an extensive coverage, availability of PlanetScope imagery can be on the order of only several minutes, with 75% of imagery collected available within 24 hours. The speed of image availability can aid first responders, given that the first 24 hours after a disaster are the most critical for saving lives.

To summarize, we find that PlanetScope data will seldom be able to completely replace more traditional satellite data for

mapping coseismic displacements such as synthetic aperture radar (SAR), Landsat and Sentinel-2, or very high resolution optical satellites, but rather complement these by filling a gap related to temporal and spatial resolution.

## 7 Code availability

The image matching code used for this study (Correlation Image AnalysiS, CIAS) is available from http://www.mn.uio.no/icemass.

## 8 Data availability

Sentinel-2 data are freely available from the ESA/EC Copernicus Sentinels Scientific Data Hub at https://scihub.copernicus.eu/ , Landsat 8 data from USGS at http://earthexplorer.usgs.gov/ . Planet data are not openly available as Planet is a commercial company. However, scientific access schemes to these data exist.

## 9 Author contribution

A.K. developed the study, did most of the analyses and wrote the paper. B.A. supported the analyses and edited the paper. J.M. helped with data acquisition, technical details to the Planet constellation and data, and edited the paper.

## 10 Competing interests

A.K. and B.A. declare no competing interests. J.M. is program manager for impact initiatives at Planet. He did in no manner influence the results or conclusions of the study.

## 11 Acknowledgements

Special thanks are due to André Stumpf, a second anonymous referee, and the editor Norman Kerle for their valuable comments and effort of reviewing and handling our paper. We also thank Seth Price from Planet for additional technical information on PlanetScope image processing. We are grateful to the providers of satellite data for this study; Planet for their cubesat data via Planet's Ambassadors Program, ESA/Copernicus for Sentinel-2 data, and USGS for Landsat 8 data. The work was funded by the European Research Council under the European Union's Seventh Framework Programme (FP/2007-2013) / ERC grant agreement no. 320816 and the ESA project Glaciers_cci (4000109873/14/I-NB).

**Figures**

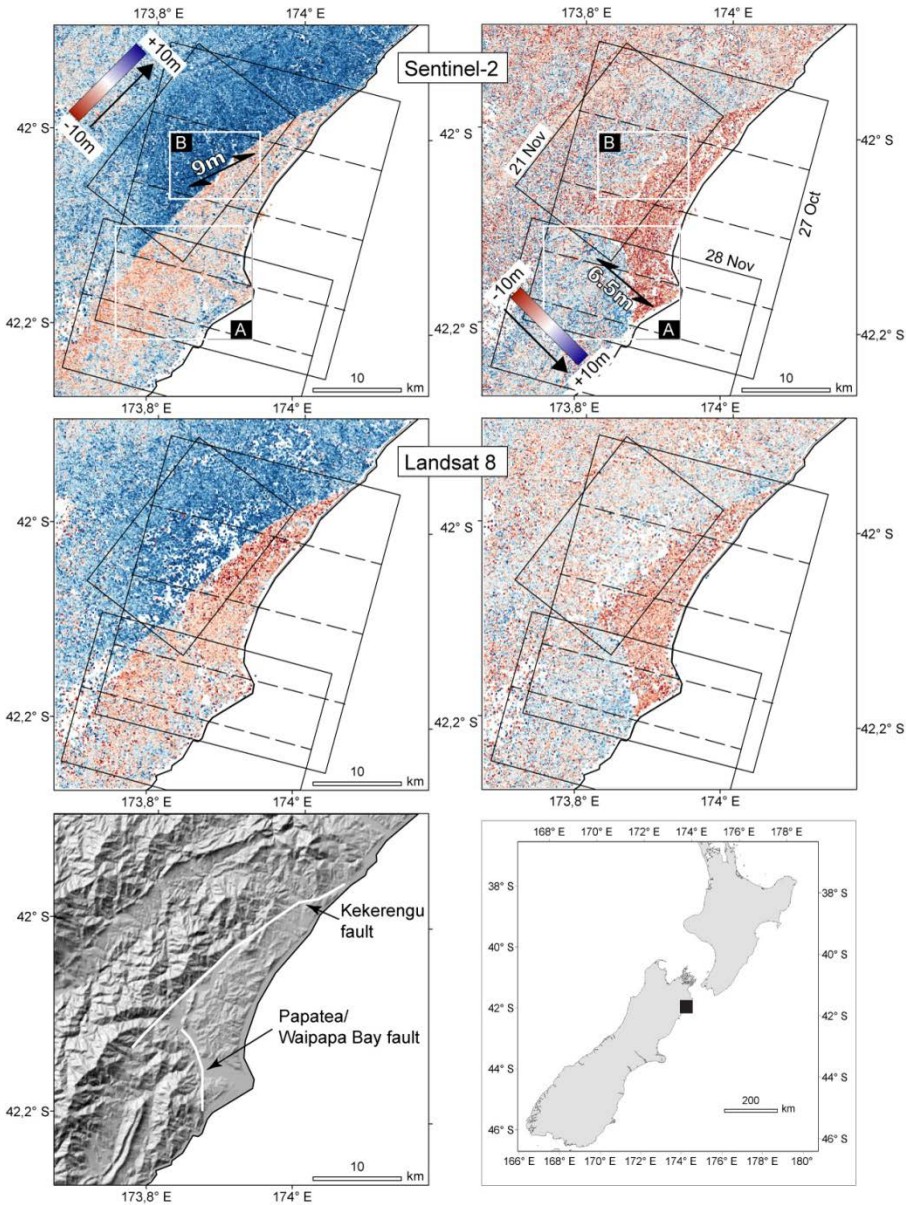

Figure 1: Sentinel 2 (3 Oct-5 Dec 2016, top row) and Landsat 8 (12 Oct-15 Dec 2016, middle row) horizontal coseismic displacements of the 14 November 2016 Kaipura, New Zealand, earthquake. Left row: SW-NE displacement component, right row: NW-SE component. Lower row, left: hillshade from the Shuttle Radar Topography Mission (SRTM); white lines schematically indicate surface ruptures from the above displacement field. Lower row, right: location of study site in New Zealand. The oblique rectangles in the upper two rows indicate the footprints of the PlanetScope images used with according dates given in the upper left panel. Inset A: Fig. 3, inset B: Fig. 4.

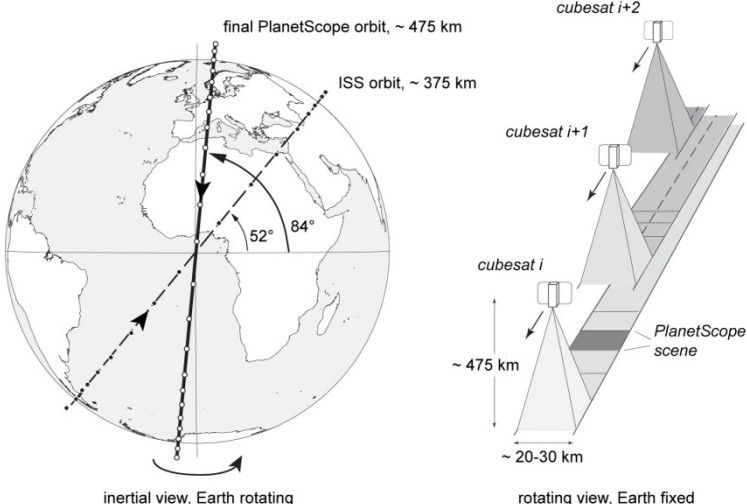

**Figure 2: left panel: final PlanetScope orbit and ISS test-bed orbit. Cubesat positions (white and black dots on the orbit) are only schematically indicated. The final PlanetScope orbit is planned to host over 100 cubesats. Right panel: scheme of complete scan of the Earth surface by successive PlanetScope cubesats (called doves) in the same orbit.**

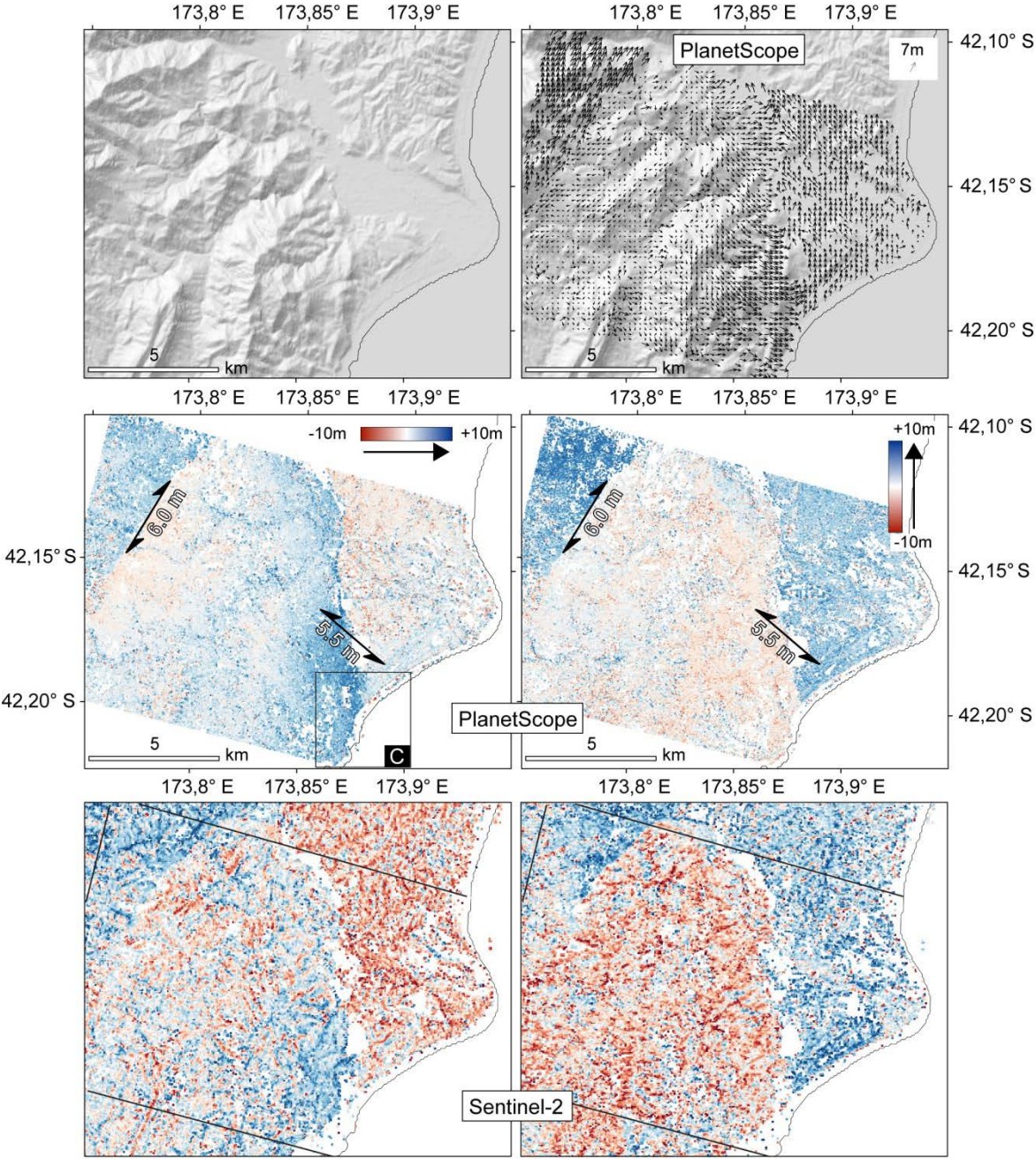

**Figure 3: Top two rows: horizontal surface displacements from PlanetScope images of 27 October – 28 November 2016. Top left: hillshade of SRTM elevation model. Top right: vectors measured originally with 20m grid spacing are resampled to 200m spacing; SRTM hillshade in background. Middle left: W-E component, right: S-N component with 20m spacing. Location of figure: A in Fig. 1. The double arrows indicate the approximate direction and the according numbers the approximate magnitude of relative displacement over ruptures. Rectangle C: Fig. 5. Lower row: as middle row but displacements from Sentinel-2 data of 3 October and 5 December 2016; same colour scale.**

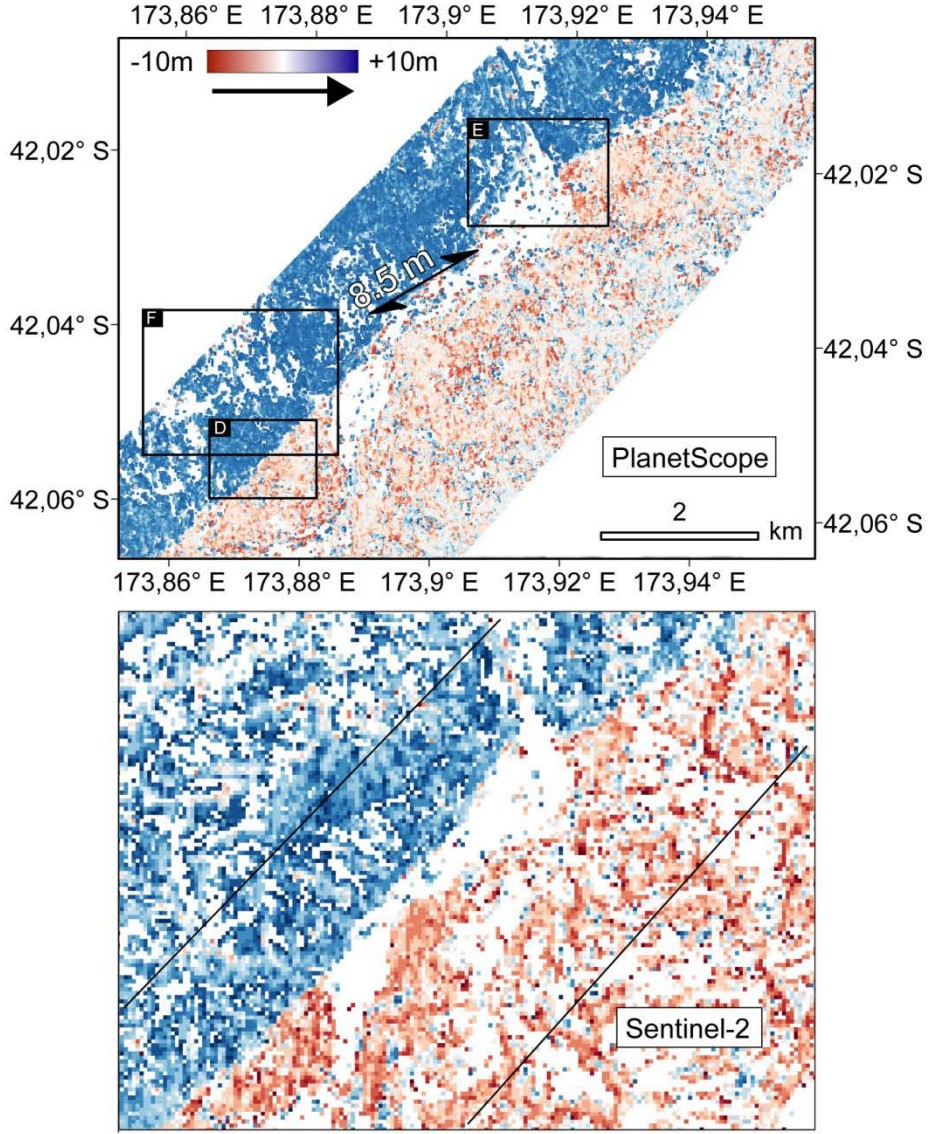

**Figure 4: Upper panel: horizontal surface displacements from PlanetScope images of 27 October – 21 November 2016, W-E component (S–N component is very similar). The double arrow indicates the approximate direction and the according number the approximate magnitude of relative displacement over the main rupture. Location of Figure: B in Fig. 1. Rectangle D: Fig. 6, rectangle E: Fig. 7, rectangle F: Fig. 8. Lower panel: as upper panel but displacements from Sentinel-2 data of 3 October and 5 December 2016; same colour scale.**

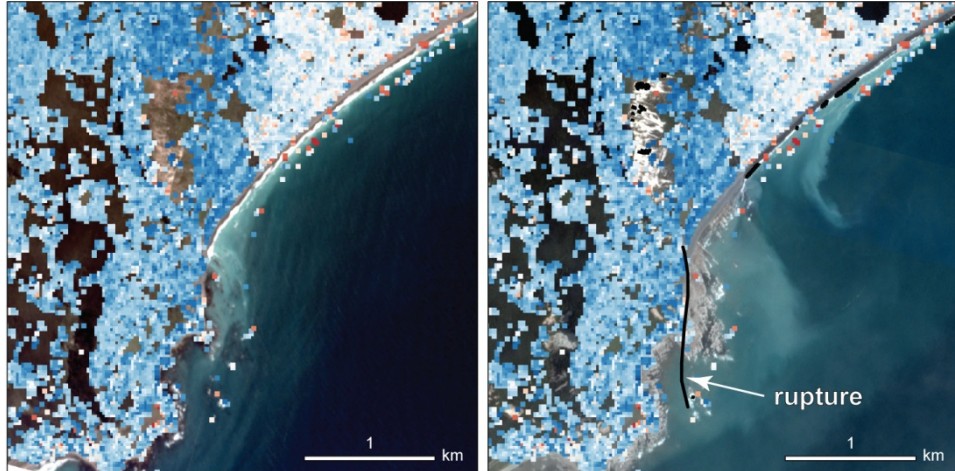

**Figure 5: Horizontal surface displacements from PlanetScope images of 27 October – 21 November 2016, W-E component. Colorscale see Fig. 3. Left background: PlanetScope image of 27 October, right background 21 November 2016. The section of uplifted seabed (right of the rupture) and the according rupture are well visible in the PlanetScope images; the rupture is indicated by a black line that was digitized from the images. Location: C in Fig. 3.**

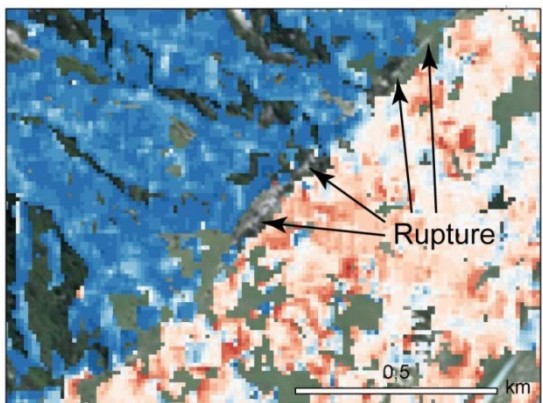

**Figure 6: Horizontal surface displacements from PlanetScope images of 27 October – 21 November 2016, W-E component. Background: PlanetScope image of 21 November with the surface rupture well visible. Location: rectangle D of Fig. 4.**

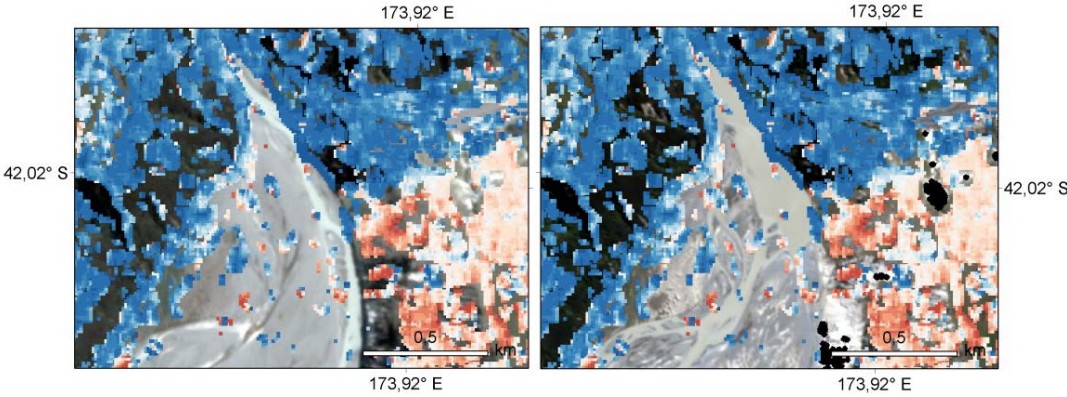

**Figure 7: Horizontal surface displacements from PlanetScope images of 27 October – 21 November 2016, W-E component. Background: PlanetScope images of 27 October (left) and 21 November (right). Clarence River was dammed up by the rupture and its course diverted. Location: rectangle E of Fig. 4.**

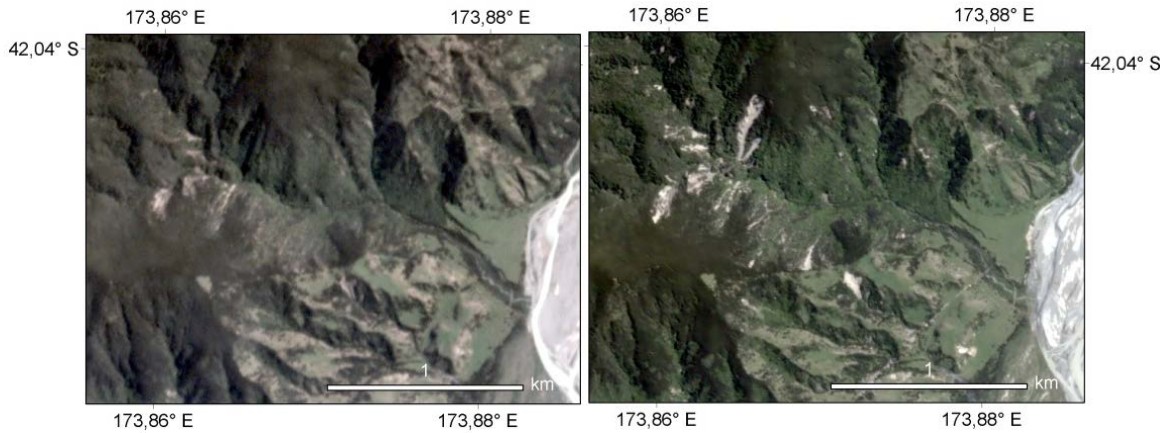

**Figure 8: PlanetScope images, left 27 October, right 21 November 2016, showing landslides by the 14 November 2016 Kaikoura earthquake. To the lower right, also the surface rupture is well visible. Location: rectangle F in Fig. 4.**

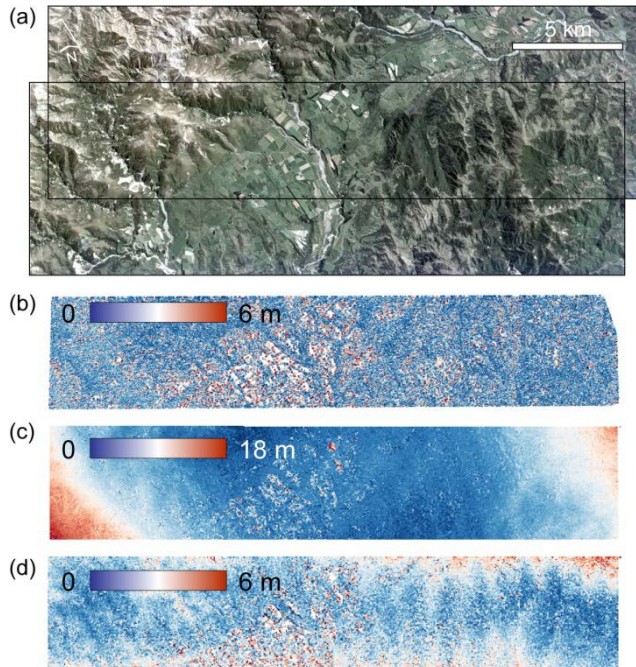

**Figure 9: Horizontal surface displacements from PlanetScope images of 20 – 25 November 2016 without surface motion expected; total magnitude of displacement. (a) the two scenes and their overlap matched in panels (b)-(d). (b) displacements between orthorectified versions. (c) displacements between the unrectified versions, when co-registered using a 1<sup>st</sup>-order polynomial. (d) like (c) but co-registered using a 2<sup>nd</sup>-oder polynomial.**

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
