# Peer review of "Coseismic displacements of the 14 November 2016 Mw7.8 Kaikoura, New Zealand, earthquake using the Planet optical cubesat constellation"

_Natural Hazards and Earth System Sciences, 2017_

## Referee Comment (RC1) · A. Stumpf (Referee) · 19 Feb 2017

In the presented study the authors evaluate the use of satellite images from Planet's cubesat constellation for measurements of co-seismic displacement resulting from the 2016 Kaikoura earthquake. Sub-pixel image correlation is used to measure the horizontal surface slip using mosaics of cubesat images over parts of Kekerengu and Papatea fault ruptures. The study includes a qualitative comparison of the derived displacement fields with results derived from pairs of Landsat-8 and Sentinel-2 images. Given the limited accessibility of ground measurements the quantitative assessment of

the derived displacement fields is focused on the variance of the measurements over areas with relatively homogeneous displacements. The authors, furthermore, include an assessment of the uncertainties through measurements of residual offset among cubesat images over stable terrain.

Given the novelty of the cubesat constellation and its potential for rapid disaster response due to very high spatial and temporal resolution, the paper provides a very interesting contribution to NHESS. Both, advantages (greater spatial detail, reduced orthorectification errors) and limitations (limited scene size) are clearly shown and discussed. The paper is well written and structured and I only have a few minor questions and suggestions which the authors may want to consider for a revision of their manuscript:

p.1: "Radar tracking methods measure the azimuth (flight direction of satellite) and range (line-of-sight) components of the displacements with, roughly, metre-accuracy (e.g., Michel et al., 1999)"

I wonder if this is still true with the availability of X-band SAR imagery and appropriate processing techniques. See for example:

Singleton, A., et al. "Evaluating sub-pixel offset techniques as an alternative to D-InSAR for monitoring episodic landslide movements in vegetated terrain." Remote Sensing of Environment 147 (2014): 133-144.

Wang, Teng, and Sigurjón Jónsson. "Improved SAR amplitude image offset measurements for deriving three-dimensional coseismic displacements." IEEE Journal of Selected Topics in Applied Earth Observations and Remote Sensing 8.7 (2015): 3271-3278.

A few questions and comments on section "2 The Planet cubesat constellation":

Since the type of corrections which can be applied depends largely on the data format it might be worth mentioning if the 'unrectified' data format can be acquired by the

general public.

I could imagine that practitioners/scientists might also be interested in some information on the life cycle of the Cubesat constellation to better evaluate the long-term perspective when relying on the provided imagery.

Could you provide any further information regarding the general co-registration accuracy and regarding the stability of the lens parameters over time according to the data provider (i.e. Planet)?

p.4: "DEMs (or DEMs for topographic phase removal within SAR interferometry) are by necessity outdated unless acquired simultaneously with image acquisition"

An example for simultaneous DEM extraction and orthorecitifcation for displacement measurements has been provided in:

Stumpf, A., Malet, J.P., Allemand, P. and Ulrich, P., 2014. Surface reconstruction and landslide displacement measurements with Pléiades satellite images. ISPRS Journal of Photogrammetry and Remote Sensing, 95, pp.1-12.

p.5: "No such type of scenes was available over the section of Fig. 1 around the earthquake date." With "No such type", do you refer to the orbit parameters (previous sentence). This could raise the impression that the scenes used over the earthquake area have been acquired with different/preliminary orbits. Please clarify.

p.5: "matching the repeat Sentinel-2 and PlanetScope data" . . . and the Landsat-8 data as well?

p.5: "no other post-processing is applied": It might be worthwhile to remind the reader at this point that L-8 and in particular S$-$2 (the global reference image is not yet used) typically comprise systematic offsets among multi-temporal acquisitions that can amount to several meters (shown for example in Kääb et al. 2016). I suppose that no post-processing was undertaken to address this issue due to the difficulty of separating image offset from ground offset in the given setting?

p.5, last paragraph: I understand that the assessment of the pointing errors of the PlanetScope data is beyond the scope of this study. However (similar to my previous comment), maybe Planet could provide some information regarding the estimated geolocation accuracy of their constellation?

p. 6: "From the standard deviation of displacements over homogenously displacing image sections we estimate a relative accuracy for individual displacements of about $\pm$ 0.4 pixels (4 m) for Sentinel-2 and for the matching window sizes, ground conditions and time interval specific to our study."

It might be worthwhile to consider also the inclusion of an equivalent quantitative analysis of the results from the Landsat-8 pair, and a quantitative comparison between the S$-$2 and L-8 results. Visually, the displacements from S$-$2 seem greater in many areas.

p.6: "we match a two-scene mosaic of 28 November 2016 with a mosaic of four scenes" Are these scenes standard orthorectified products as provided by Planet or have you processed 'unrectified' imagery particularly for this study? Given the variable ground sampling distance of the constellation, was it necessary to perform resampling before the matching?

p. 7: "Again, the displacement from PlanetScope data agrees well within error bounds with the Sentinel-2 results of 9 m." To further quantify the relative uncertainty of the measurements, would it be possible to include a quantitative comparison between the S-2 and PlanetScope results (e.g. a figure showing the difference of the two)? Possible offsets between the two products could be accounted by aligning the fault traces for example.

Section 4.3 Stable ground test: Could you provide any further information regarding the co-registration / orthorectification procedure used by Planet? I.e. if the lens model is re-estimated on an image-to-image base through matching of homologous points, ground displacement could propagate into the re-estimation of the lens model.

[Figure]

p.9:"enables relative measurement accuracies of as low as ±0.2 pixels (∼ ±0.6 m) for individual displacements,..." and "Finally, the above matching accuracy of on the order of ±1 m will prevent detecting small (coseismic) displacements."

It seems not entirely clear where these number comes from. In section 4.2 the estimate is "± 0.7 pixels (2 m)" and in section 4.3 the "variability of the individual displacements, is around 1.9 m" over stable ground. Similarly the "±0.2 pixels (∼ ±0.6 m)" in the abstract seems a bit optimistic. Please clarify.

Figure 4: The letters for the insets differ between the figure, and the caption and the main text.

Figures 3-7: To better illustrate the additional detail provided from the PlantScope-based displacement fields it might be helpful to provide (at least for 1 or 2 figures) as side-by-side view with the corresponding S-2-based results.

---

## Referee Comment (RC2) · Anonymous Referee #2 · 21 Mar 2017

This paper is an interesting contribution to the use of the new satellite image data provided by PlanteScope cubesat constellation to measure earth surface displacement by optical correlation. On the example of a very recent earthquakes in New Zealand, they show the potential and limitations of such data and compare them to Landsat 8 and Sentinel 2 results. The results are well described and the figures are clear. The point that can be improved is the organization of the end of the paper between the results-discussion part and the conclusion-perspective part. I suggest to write a results part followed by a discussion part, and to extract only the main results of the paper in the conclusion without adding new ideas.

---

## Author Comment (AC1) · 28 Mar 2017

**Natural Hazards and Earth System Sciences        nhess-2017-30**

**Coseismic displacements of the 14 November 2016 Mw7.8 Kaikoura, New Zealand, earthquake using an optical cubesat constellation**

**Andreas Kääb et al.**

**General response**

We would like to thank the two referees for their constructive and thoughtful reviews which certainly helped to improve the paper!

In response to the reviews we changed the text at many places throughout the manuscript, added information, and restructured the results, discussion and conclusions.

We hope to have that way addressed the referee comments in a satisfactory way.

**Response to individual referees**

Referee #1, Andre Stumpf

*In the presented study the authors evaluate the use of satellite images from Planet's cubesat constellation for measurements of co-seismic displacement resulting from the 2016 Kaikoura earthquake. Sub-pixel image correlation is used to measure the horizontal surface slip using mosaics of cubesat images over parts of Kekerengu and Papatea fault ruptures. The study includes a qualitative comparison of the derived displacement fields with results derived from pairs of Landsat-8 and Sentinel-2 images. Given the limited accessibility of ground measurements the quantitative assessment of the derived displacement fields is focused on the variance of the measurements over areas with relatively homogeneous displacements. The authors, furthermore, include an assessment of the uncertainties through measurements of residual offset among cubesat images over stable terrain.*

*Given the novelty of the cubesat constellation and its potential for rapid disaster response due to very high spatial and temporal resolution, the paper provides a very interesting contribution to NHESS. Both, advantages (greater spatial detail, reduced orthorectification errors) and limitations (limited scene size) are clearly shown and discussed.*

*The paper is well written and structured and I only have a few minor questions and suggestions which the authors may want to consider for a revision of their manuscript:*

*p.1: "Radar tracking methods measure the azimuth (flight direction of satellite) and range (line-of-sight) components of the displacements with, roughly, metre-accuracy (e.g., Michel et al., 1999)"*

*I wonder if this is still true with the availability of X-band SAR imagery and appropriate processing techniques. See for example:*
*Singleton, A., et al. "Evaluating sub-pixel offset techniques as an alternative to D-InSAR for monitoring episodic landslide movements in vegetated terrain." Remote Sensing of Environment 147 (2014): 133-144.*
*Wang, Teng, and Sigurjón Jónsson. "Improved SAR amplitude image offset measurements for deriving three-dimensional coseismic displacements." IEEE Journal of Selected Topics in Applied Earth Observations and Remote Sensing 8.7 (2015): 3271-3278.*

We think that 'metre-accuracy' is still true as an order of magnitude, but include both references and specify that better accuracies can be obtained in particular for strong artificial or natural reflectors.

*A few questions and comments on section "2 The Planet cubesat constellation":*
*Since the type of corrections which can be applied depends largely on the data format it might be worth mentioning if the 'unrectified' data format can be acquired by the general public.*

Yes, available to the public. Now specified.

*I could imagine that practitioners/scientists might also be interested in some information on the life cycle of the Cubesat constellation to better evaluate the long-term perspective when relying on the provided imagery.*

Now specified. We anticipate that the Doves in sun synchronous orbit will function for 3-5 years. Planet plans to keep the constellation complete by continuously supplying new satellites.

*Could you provide any further information regarding the general co-registration accuracy and regarding the stability of the lens parameters over time according to the data provider (i.e. Planet)?*

Information added to the extent it is available.

*p.4: "DEMs (or DEMs for topographic phase removal within SAR interferometry) are by necessity outdated unless acquired simultaneously with image acquisition"*
*An example for simultaneous DEM extraction and orthorecitifcation for displacement measurements has been provided in:*
*Stumpf, A., Malet, J.P., Allemand, P. and Ulrich, P., 2014. Surface reconstruction and landslide displacement measurements with Pléiades satellite images. ISPRS Journal of Photogrammetry and Remote Sensing, 95, pp.1-12.*

True. Your example was now incorporated.

*p.5: "No such type of scenes was available over the section of Fig. 1 around the earthquake date." With "No such type", do you refer to the orbit parameters (previous sentence). This could raise the impression that the scenes used over the earthquake area have been acquired with different/preliminary orbits. Please clarify.*

You are right. The entire paragraph has been clarified in this direction.

*p.5: "matching the repeat Sentinel-2 and PlanetScope data"*
*... and the Landsat-8 data as well?*

True. Corrected.

*p.5: "no other post-processing is applied": It might be worthwhile to remind the reader at this point that L-8 and in particular S−2 (the global reference image is not yet used) typically comprise systematic offsets among multi-temporal acquisitions that can amount to several meters (shown for example in Kääb et al. 2016). I suppose that no post-processing was undertaken to address this issue due to the difficulty of separating image offset from ground offset in the given setting?*

Yes, we specified now that systematic offset patterns have not been studied and corrected, and that offsets are thus relative only.

*p.5, last paragraph: I understand that the assessment of the pointing errors of the PlanetScope data is beyond the scope of this study. However (similar to my previous comment), maybe Planet could provide some information regarding the estimated geolocation accuracy of their constellation?*

Now added.

*p. 6: "From the standard deviation of displacements over homogenously displacing image sections we estimate a relative accuracy for individual displacements of about ±0.4 pixels (4 m) for Sentinel-2 and for the matching window sizes, ground conditions and time interval specific to our study."*

*It might be worthwhile to consider also the inclusion of an equivalent quantitative analysis of the results from the Landsat-8 pair, and a quantitative comparison between the S−2 and L-8 results. Visually, the displacements from S−2 seem greater in many areas.*

We added the relative accuracy for Landsat 8, and write a short statement on the S2 – L8 differences (see figure below). We don't however add the below figure to the main manuscript as we think it is of too marginal importance for the main focus of the paper – the Planet constellation.

[Figure]

*Left: S2-L8 differences SW-NE, right: NW-SE, i.e. differences between the top two rows of Fig. 1 (color scale as in Fig. 1).*

*p.6: "we match a two-scene mosaic of 28 November 2016 with a mosaic of four scenes" Are these scenes standard orthorectified products as provided by Planet or have you processed 'unrectified' imagery particularly for this study? Given the variable ground sampling distance of the constellation, was it necessary to perform resampling before the matching?*

We specified now that we used standard Planet ortho-products without any corrections and adjustments. No resampling was necessary as all original images used had the same resolution (by chance).

*p. 7: "Again, the displacement from PlanetScope data agrees well within error bounds with the Sentinel-2 results of 9 m." To further quantify the relative uncertainty of the measurements, would it be possible to include a quantitative comparison between the S-2 and PlanetScope results (e.g. a figure showing the difference of the two)? Possible offsets between the two products could be accounted by aligning the fault traces for example.*

After experimenting with several visualizations, we prefer to show the Sentinel-2 derived displacements together with the PlanetScope ones for the same image sections in Figs. 3 and 4. This gives a more complete impression than a map of differences between both measurements. Summary statistics of the differences are added in the text.

*Section 4.3 Stable ground test: Could you provide any further information regarding the co-registration / orthorectification procedure used by Planet? I.e. if the lens model is re-estimated on an image-to-image base through matching of homologous points, ground displacement could propagate into the re-estimation of the lens model.*

Information added to the extent it is available.

*p.9:"enables relative measurement accuracies of as low as ±0.2 pixels ( ~±0.6 m) for individual displacements,..." and "Finally, the above matching accuracy of on the order of ± 1 m will prevent detecting small (coseismic) displacements."*

*It seems not entirely clear where these number comes from. In section 4.2 the estimate is "±0.7 pixels (2 m)" and in section 4.3 the "variability of the individual displacements, is around 1.9 m" over stable ground. Similarly the "±0.2 pixels (∼±0.6 m)" in the abstract seems a bit optimistic. Please clarify.*

These numbers stem from three different tests. Now summarized and clarified in the conclusions, and a range for accuracies given in the abstract covering all three tests, not only the most optimistic one (or actually the most realistic one for the final Planet constellation?).

*Figure 4: The letters for the insets differ between the figure, and the caption and the main text.*

Repaired.

*Figures 3-7: To better illustrate the additional detail provided from the PlantScope-based displacement fields it might be helpful to provide (at least for 1 or 2 figures) as side-by-side view with the corresponding S-2-based results.*

Done, see also above response to the comment about p.7.
* * *
Referee #2
* * *
*This paper is an interesting contribution to the use of the new satellite image data provided by PlanteScope cubesat constellation to measure earth surface displacement by optical correlation. On the example of a very recent earthquakes in New Zealand, they show the potential and limitations of such data and compare them to Landsat 8 and Sentinel 2 results. The results are well described and the figures are clear. The point that can be improved is the organization of the end of the paper between the results-discussion part and the conclusion-perspective part. I suggest to write a results part followed by a discussion part, and to extract only the main results of the paper in the conclusion without adding new ideas.*

We restructured the paper by separating results and discussion. We also restructured the conclusions a bit, but preferred to keep most of the few short perspectives.

---

## Author Comment (AC2) · 28 Mar 2017

The comment was uploaded in the form of a supplement:
http://www.nat-hazards-earth-syst-sci-discuss.net/nhess-2017-30/nhess-2017-30-AC2-supplement.pdf